# The Stem-Loop I of Senecavirus A IRES Is Essential for Cap-Independent Translation Activity and Virus Recovery

**DOI:** 10.3390/v13112159

**Published:** 2021-10-26

**Authors:** Nana Wang, Haiwei Wang, Jiabao Shi, Chen Li, Xinran Liu, Junhao Fan, Chao Sun, Craig E. Cameron, Hong Qi, Li Yu

**Affiliations:** 1State Key Laboratory of Veterinary Biotechnology, Harbin Veterinary Research Institute, Chinese Academy of Agricultural Sciences, Harbin 150069, China; wangnana123_2@163.com (N.W.); wanghaiwei@caas.cn (H.W.); shijiabao1001@163.com (J.S.); carolindia@126.com (C.L.); ffanjh@163.com (J.F.); sunchao@caas.cn (C.S.); 2Department of Chemistry, The Pennsylvania State University, University Park, PA 16802, USA; xinranliu603@gmail.com; 3Department of Microbiology and Immunology, University of North Carolina School of Medicine, Chapel Hill, NC 27516, USA; craig_cameron@med.unc.edu; 4Key Laboratory of Urban Water Resource and Environment, Harbin Institute of Technology, School of Environment, Harbin 150090, China

**Keywords:** Senecavirus A, IRES, stem-loop I, translation, viral replication, picornavirus

## Abstract

Senecavirus A (SVA) is a picornavirus that causes vesicular disease in swine and the only member of the *Senecavirus* genus. Like in all members of *Picornaviridae*, the 5′ untranslated region (5’UTR) of SVA contains an internal ribosome entry site (IRES) that initiates cap-independent translation. For example, the replacement of the IRES of foot-and-mouth disease virus (FMDV) with its relative bovine rhinitis B virus (BRBV) affects the viral translation efficiency and virulence. Structurally, the IRES from SVA resembles that of hepatitis C virus (HCV), a flavivirus. Given the roles of the IRES in cap-independent translation for picornaviruses, we sought to functionally characterize the IRES of this genus by studying chimeric viruses generated by exchanging the native SVA IRES with that of HCV either entirely or individual domains. First, the results showed that a chimeric SVA virus harboring the IRES from HCV, H-SVA, is viable and replicated normally in rodent-derived BHK-21 cells but displays replication defects in porcine-derived ST cells. In the generation of chimeric viruses in which domain-specific elements from SVA were replaced with those of HCV, we identified an essential role for the stem-loop I element for IRES activity and recombinant virus recovery. Furthermore, a series of stem-loop I mutants allowed us to functionally characterize discrete IRES regions and correlate impaired IRES activities, using reporter systems with our inability to recover recombinant viruses in two different cell types. Interestingly, mutant viruses harboring partially defective IRES were viable. However, no discernable replication differences were observed, relative to the wild-type virus, suggesting the cooperation of additional factors, such as intermolecular viral RNA interactions, act in concert in regulating IRES-dependent translation during infection. Altogether, we found that the stem-loop I of SVA is an essential element for IRES-dependent translation activity and viral replication.

## 1. Introduction

Seneca Valley virus, also known as Senecavirus A (SVA), is the only member of the genus *Senecavirus* in the family *Picornaviridae*. SVA causes vesicular lesions in swine and acute death in neonatal piglets, resulting in significant economic losses for the porcine industry [1]. Since 2014, increased SVA infection has been reported in many countries, including Brazil, the United States, Columbia, Thailand, and China [2,3,4,5,6,7]. The SVA genome is a positive-sense, single-stranded RNA composed of a single open reading frame (ORF) flanked by two untranslated regions (5’ UTR and 3’ UTR). The SVA genome can also act as a template for the translation of the viral proteins. However, unlike eukaryotic RNA, a structured RNA in the 5’ UTR called type IV internal ribosome entry site (IRES) [8] is used to recruit distinct cellular proteins and thus, initiates translation in a cap-independent manner [9,10,11].

It was shown that the IRES element of certain picornaviruses, including SVA, is closely related to the IRES found in hepatitis C virus (HCV), which belongs to the *Hepacivirus* genus within the *Flaviviridae* family [12]. At the genetic level, the IRES sequences of SVA and HCV exhibit an overall identity of 52% and share similar secondary structures [13]. The SVA IRES spans about 300 nucleotides containing two major domains: domain II and domain III. Domain III contains a number of different stem-loop structures and a pseudoknot referred to as domain IIIf or IV [14,15]. The pseudoknot region is formed by two base-paired stem regions (stem I and II) and, although the sequences of the stems are not particularly well-conserved, the presence of the structure is highly conserved with compensatory base changes to preserve the base pairing [16,17]. A previous study found that the secondary structure of SVA IRES is more similar to classical swine fever virus (CSFV) IRES than to the HCV IRES because of the presence of domain IIId2 [18,19]. However, no direct correlation was observed between domain IIId2 and IRES functions [20,21,22]. A previous study of SVA IRES found that a 55-nucleotide sequence is required for optimal SVA IRES activity without the requirement for the activity of eIF4F factors to function as for other minimal sequences [17].

Due to the essential role of IRES in cap-independent translation, this RNA element provides a functional tool for the study of replication and virulence of RNA viruses. Indeed, previous studies demonstrated that recombinant polioviruses under the control of the IRES of human rhinovirus 2 or HCV were incapable of accumulating or causing disease in the brain and spinal cord of mice, indicating a role in pathogenicity [23]. Moreover, the replacement of IRES domains III or IV with the corresponding domain of bovine rhinitis B virus (BRBV) impairs the replication of foot-and-mouth disease virus (FMDV) in porcine-derived cells [24]. In addition, we have recently shown that a single nucleotide cytosine-to-guanine substitution at position 351 of the IRES endows FMDV with temperature-sensitive and attenuation phenotypes, leading to the generation of highly attenuated mutant with an immune protective capacity [25], further highlighting the key roles of the picornaviral IRES in viral replication. 

Here, we have explored functionally important structures in the SVA IRES by constructing and characterizing chimeric viruses between HCV and SVA, site-directed mutagenesis, dual-luciferase reporter assay, RT-qPCR assay, and virus recovery study. It was found that viral replication of the SVA chimeric virus containing the IRES of HCV was severely impaired in porcine-derived cells. Further, the results identified that IRES-dependent translation activity and the recovery of viable recombinant SVA depend on the structure of complementary base pairing of stem-loop I in the SVA IRES. Domain-specific replacement with HCV counterparts and point mutations disrupting this structure in stem-loop I (SL1) reduced IRES function and negatively affected recovery of these viral mutants. Thus, the data presented here reveal a novel determinant of SVA replication in vitro, offering key information for the design of vaccines, using attenuated viruses.

## 2. Materials and Methods

### 2.1. Cells and Viruses

Baby hamster kidney (BHK-21; ATCC CCL-10) cells and porcine-derived swine testicular (ST) cells were maintained in Dulbecco’s modified Eagle’s medium (DMEM; Invitrogen, Carlsbad, CA, USA) containing 10% fetal bovine serum (FBS; HyClone, Logan, UT, USA) and 2 mM L-glutamine at 37 °C under 5% CO_2_. The wild-type SVA/HLJ/CHA/2016 virus (GenBank accession number KY419132) was isolated from finishing pigs on a farm in Heilongjiang Province in northeast China in 2016, as described previously [19]. The virus stocks were expanded in BHK-21 cells.

### 2.2. Construction of SVA/HCV IRES Chimeras

SVA mutants with complete HCV IRES replacement or different IRES domains were constructed by standard overlapping PCR, using the primers listed in Table 1. To construct H-SVA, primer pairs 1 + 4 and 7 + 2 were used to generate the 5′ UTR ΔIRES and partial L gene sequences of SVA from the infectious clone pSVA-16 plasmid. Simultaneously, primer pairs 5 + 6 were used to amplify HCV IRES from the synthetic HCV IRES sequences. The resulting PCR products were used as templates to amplify overlapping products, using primer pairs 1 + 2. The purified PCR product and pSVA-16 plasmid were subsequently digested with restriction endonucleases *Nhe*I and *Sac*II (NEB, Beijing, China) and ligated using T4 DNA ligase (NEB). The final clone, named pH-SVA, was identified by sequencing. A similar approach was used to construct recombinant plasmids containing HCV IRES, using the primers shown in Table 1. These were named H-II-SVA, H-III-SVA, H-IV-SVA, H-II/III-SVA, H-II/III-SS1-SVA, H-III/IV-SVA, and H-III/IV-HS1-SVA.

Additionally, we constructed 15 SVA mutants for site-directed mutagenesis of stem-loop I, using fusion PCR. As an example of the process, the construction of the L-DelCU/UU mutant was as follows: a 450 bp fragment was firstly amplified from pSVA-16 plasmid using primer pair 1 + 16, which included the restriction site of *Nhe*I and the deletion of base CU at 442 and 443 on the stem-loop I. Next, a 200 bp fragment including the deletion of 442 and 443 base CU and 646 and 647 base UU on the stem-loop I was amplified, using primer pair 17 + 18. Similarly, a 400 bp fragment with the deletion of 646 and 647 bases of UU and *Sac*II restriction sites on the stem-loop I was amplified, using primer pair 19 + 2. Finally, the resulting PCR products were used as templates to amplify overlapping products, using the primer pair 1 + 2 for fusion PCR to obtain the 5′ UTR fragment of SVA IRES with all the bases missing on the stem-loop I. The purified PCR fragment and the pSVA-16 plasmid were digested with restriction endonucleases *Nhe*I and *Sac*II, and ligated using T4 DNA Ligase, producing L-del CU/UU. The same fusion PCR method was used to construct the other 14 SVA mutants for site-directed mutagenesis of the stem-loop I, using the primers listed in Table 1.

### 2.3. Construction of a Subgenomic Replicon Containing Dual-Luciferase Reporter Gene 

To construct the dual-SVA reporter plasmids, primer pairs in Table 2 were used to amplify the IRES sequence containing a 55 nucleotide coding sequence from pSVA-16; the sequence contained the *Bgl*II and *Nco*I restriction sites with small repeats from PGL-3 vector at both ends, producing SVA-IRES-55nt/BglII/NcoI. The RLuc gene was inserted into a PGL-3 vector containing a T7 promoter (containing *Bgl*II restriction site) and the Fluc gene was inserted into PGL-3 (containing *Nco*I restriction site). After digestion with restriction endonucleases and DNA fragment purification, SVA-IRES-55nt/BglII/NcoI was fused to this segment and named Dual-SVA. A similar strategy was used to create the reporter plasmids for the other mutants.

### 2.4. Recovery of Recombinant Viruses

BHK-21 cells seeded in 6-well plates were transfected with the infectious clone plasmids using Lipofectamine 3000 following the manufacturer’s instructions (Life Technologies, Carlsbad, CA, USA). The cytopathic effect (CPE) was monitored daily after infection and recombinant viruses were harvested when significant levels of CPE were observed. The recovered viruses were passaged 10 times in BHK-21 cells, and the stability of the introduced mutations was confirmed by sequencing.

### 2.5. Luciferase Assay

The in vitro transcribed replicon RNA was transfected into monolayers of BHK-21 and ST cells, using Effectene Transfection Reagent (Qiagen, Germany). At 12 h (h) post transfection, the growth medium was removed from the dishes, and the cells were washed gently with 2 mL PBS. Fluc and Rluc activities were measured in cell lysates, using a Dual-Luciferase^®^ Reporter Assay System (Promega, Madison, WI, USA) and a microplate reader (Bio-Tek, Waltham, MA, USA).

### 2.6. TCID_50_ Assay and Growth Curve 

Tenfold serial dilutions of the virus were prepared in 96-well round-bottom plates in DMEM. Dilutions were performed in octuplicate, and 50 μL of the dilution was transferred to 10^4^ BHK-21 cells seeded in a volume 100 μL of DMEM with 2% FBS. After 3 days, 50% tissue culture infective dose (TCID_50_) values were determined by the Reed-Muench formula.

To determine viral replication kinetics, growth experiments in BHK-21 cells and ST cells were performed as follows. First, cell monolayers in 6-well tissue culture plates were washed with phosphate-buffered saline (PBS) and inoculated with the different viruses at a multiplicity of infection (MOI; equal to PFU number/cell) of 0.01. The plates were incubated for 1 h at 37 °C. Then, the cells were washed three times with PBS to remove unbound virus particles and overlayed with DMEM supplemented with 2% FBS. The infected cells were incubated at 37 °C and harvested at different times. The plates were subjected to three consecutive freeze-thaw cycles, and cell debris was removed by centrifugation. The viral titers of the supernatants were determined by a TCID_50_ assay. Mean values and standard deviations were calculated from three independent experiments.

### 2.7. RNA Extraction and Real-Time RT-qPCR

Total RNA was extracted from SVA-infected cells using a Simply P total RNA extraction kit (BioFlux, Redwood, CA, USA), according to the manufacturer’s instructions, and total RNA was used for cDNA synthesis with PrimeScript reverse transcriptase (TaKaRa, Kyoto, Japan). cDNA quantification was performed by using an Mx3005P instrument (Agilent Technologies, Palo Alto, CA, USA) as described previously [26]. The mean values and standard deviations were derived from independent, triplicate measurements. 

### 2.8. Statistical Analysis

Data handling and analysis, and graphic representation were performed, using Prism (version 6.0) software (GraphPad Software, San Diego, CA, USA). Significant differences were determined using Student’s t test. *p* values > 0.05 were considered not significant (NS).

## 3. Results

### 3.1. HCV IRES Replacement Conferred SVA Substantial Replicative Disadvantages in Porcine-Derived ST Cells

In silico predictions of the RNA secondary structure of the SVA IRES and the HCV IRES revealed shared similarities (Figure 1A). First, we sought to determine the role of the natural IRES element on the replication capacity of SVA. We constructed a chimeric virus in which the IRES of SVA was replaced in its entirety with that of HCV using the SVA infectious clone as the backbone. The chimeric virus was successfully rescued in culture and designated as H-SVA. To evaluate the effect of this IRES replacement on virus replication and infectivity, the growth kinetics of H-SVA were compared to those of the wild-type (WT) SVA virus in two different cell lines. These experiments showed similar growth kinetics between the chimeric H-SVA virus and the WT SVA in rodent-derived BHK-21 cells, indicating that the IRES from HCV could functionally replace that of SVA without noticeable effects on its growth in BHK-21 cells (Figure 1C). However, in ST cells derived from porcine, the natural host of SVA, the chimeric virus displayed slower kinetics and overall lower titers relative to WT virus (Figure 1C). These results demonstrated that the capacity of the HCV IRES to replace the natural SVA IRES functionally is cell type dependent.

In addition to viral growth assays, we quantified the viral RNA from each sample harvested throughout the experiments, using a quantitative real time-PCR (qRT-PCR). As shown in Figure 1D, the total RNA yields were similar between the chimeric H-SVA and the WT SVA throughout the course of infection in BHK-21 cells. However, total viral RNA copies for the H-SVA mutant were significantly reduced, compared to those of the WT virus in porcine-derived ST cells. These results confirmed the cell type-specific effects of IRES swapping on SVA replication and suggested an overall reduction in viral RNA synthesis in porcine-derived cells for the chimeric virus.

To determine the effect of complete IRES replacement on SVA translation initiation efficiency, we constructed dual Firefly luciferase (FLuc) and Renilla luciferase (RLuc) reporter plasmids to quantify the IRES activity of both the WT and the chimeric H-SVA viruses. It was reported that optimal SVA IRES activity requires about 55 nucleotides of coding sequence [27]. Therefore, either the WT SVA IRES (Dual-SVA) or the HCV IRES (Dual-H-SVA) sequences followed by the first 55 nucleotides of the SVA coding sequence were inserted between the Rluc and Fluc open reading frames of a bicistronic pGL3-Rluc/IRES/Fluc reporter plasmid under the control of a T7 promoter (Figure 1B). The resulting plasmids were transfected into BHK-21 cells or ST cells engineered to express the bacteriophage T7 RNA polymerase, and then the ratio of FLuc to RLuc was used to quantify IRES translational activity (Figure 1E). These experiments showed that the translation initiation efficiency for the chimeric Dual-H-SVA was significantly reduced, compared to the natural IRES in both BHK-21 and ST cells.

Collectively, these results showed that replacement of SVA IRES with that of HCV reduces SVA replication capacity in porcine-derived cells, likely by decreasing the translational initiation capacity of the IRES, which in turn, impacts the level of viral RNA synthesis in porcine cells.

### 3.2. Domain IV Is Essential for the IRES Translation–Initiation Activity and Recovery of Viable Virus 

To identify the IRES domain responsible for the impaired growth phenotype seen for the chimeric SVA in porcine-derived cells, we constructed a series of domain-specific chimeric SVA mutants, where each domain of the SVA IRES was individually replaced by its counterpart from the structurally similar HCV IRES (Figure 2A). All of the mutant viruses were successfully rescued, except for the mutant H-IV-SVA carrying the domain IV from HCV IRES. Subsequently, the growth kinetics of H-II-SVA carrying the domain II from HCV IRES and H-III-SVA carrying the domain III from HCV IRES mutants were compared to that of the WT virus by infecting BHK-21 and ST cells with the different viruses at an MOI of 0.01 and quantifying virus titers at multiple times post-inoculation in both BHK-21 and ST cells. These experiments showed that H-II-SVA and H-III-SVA replicated as efficiently as WT SVA in BHK-21 cells and ST cells (Figure 2B), indicating that the domains II and III of HCV can functionally replace those of SVA and are likely not the determinant factors in the impairment seen before in the H-SVA chimera.

As performed previously, we also quantified the levels of viral RNA for the different viruses, using qRT-PCR. Both chimeric mutants, H-II-SVA and H-III-SVA, showed similar levels of viral RNA to the WT SVA in both BHK-21 and ST cells, further indicating that domains II and III of SVA IRES can be independently replaced by the corresponding domains from HCV IRES without significant effects on viral RNA replication dynamics (Figure 2C).

In order to determine the effect of independent replacement of each domain of SVA IRES on its translation initiation efficiency, we constructed the plasmids of Dual-H-II-SVA, Dual-H-III-SVA and Dual-H-IV-SVA and quantified dual-luciferase activities after transfection of BHK-21 cells and ST cells. As shown in Figure 2D, independent replacement of the IRES domain IV of the SVA had the lowest translational activity of the IRES. Although the generation of a domain IV chimeric virus was unsuccessful, this observation would strongly suggest a key role for this domain in regulating viral translation and explain our inability to recover the recombinant H-IV-SVA mutant virus.

### 3.3. The Stem-Loop I within Domain IV of SVA IRES Is Required for Recovery of Virus 

Secondary structure analyses of SVA IRES showed that the nucleotide sequence between domains II and III is complementary to a sequence within domain IV, forming the complementary structure stem-loop I (SL I). We speculated that the replacement of domain IV destroyed the base pairing of SVA IRES secondary structure (Figure 3A), explaining the dramatic effects seen in chimeric mutants. To test our assumption, combination mutants including the IRES domain II, domain III and domain IV of SVA were replaced with their counterparts from the HCV IRES. In mutants where SVA IRES domain II, domain III and half of stem-loop I were all replaced with their corresponding regions of HCV, they were designated as H-II/III-SVA (Figure 3B). The nucleotides located on the top of the mutant IRES SL I were from the HCV and the others, which adjacent to stem II, remained unchanged. As expected, we were unable to rescue a viable mutant. In turn, an additional mutant, H-II/III-SS1-SVA, in which the nucleotides of both sides of SL I were from the SVA IRES, could be successfully rescued. The same strategy was applied for the construction of two SL I mutants within domains III and IV. Unexpectedly, no viable virus could be rescued from the two mutants H-III/IV-SVA and H-III/IV-HSI-SVA (Figure 3B). These results suggest that IRES stem-loop I is critical for the recovery of viable SVA mutants, highlighting an unprecedented key role for this region in SVA replication, likely by being indispensable for viral translation.

In addition, dual-luciferase reporter plasmids Dual-H-II/III-SVA, Dual-H-II/III-SS1-SVA, Dual-H-III/IV-SVA and Dual-H-III/IV-HS1-SVA were constructed and transfected into BHK-21 cells and ST cells to investigate their effects on IRES translation initiation efficiency. As expected, given our previous data, the translation initiation efficiency of Dual-H-II/III-SVA, Dual-H-III/IV-SVA and Dual-H-III/IV-SVA, in which the base pairing of SL I was completely disrupted, was significantly reduced when compared with the Dual-SVA control (Figure 3C). In contrast, the replicon Dual-H-II/III-SSI-SVA retaining the base pairing of SL I had similar translation initiation efficiency with that of SAV WT IRES. Thus, the stem-loop I of SVA is crucial for the translational activity of the IRES, and manipulation of this region often results in nonviable viruses.

### 3.4. Mutations in Stem-Loop I Affect the Translation Initiation Activity of IRES in Different Cell Lines

Given that SVA IRES stem-loop I is essential for viral replication, we focused on this region to further define the relationship between its structure and function through genetic mutations in BHK-21 cells. We removed nucleotides in the loop in three ways: deletion of the entire loop (mut1), deletion of nucleotides CU (nucleotide at positions 442 and 647) from the top loop (mut 2), and nucleotides UU (nucleotide at position 443 and 646) from the bottom loop (mut 3). The three mutants were analyzed, using the dual-luciferase reporter construct in BHK-21 cells as previously described. All of the IRES activities of the three mutants were significantly impaired, showing an overall reduction in activity of about 30% compared to that of the SVA WT IRES. We next created a set of mutants with inversion mutations. Inverting the nucleotides at positions 442 and 647 from CU to UC (mut4) retained ~65% of WT IRES activity, suggesting that these mutations had less of an effect on IRES activity than the deletion mutants. Notably, inversion mutants on the loop into base-paired nucleotides—including nucleotides at positions 442 and 647 from CU to either CG (mut5) or AU (mut6), 443 and 646 from UU to either AU (mut7) or UA (mut8)—also led to a reduction of 20–40% activity relative to WT IRES (Figure 4A). In sharp contrast, an even stronger decrease in IRES activity (to about 20% of WT IRES levels) was observed for the two mutants in which we deleted paired bases at positions 444 and 645 (CG) (mut10) and two paired bases (^444^CC^445^-^645^GG^646^) (mut9) located adjacent to the loop on the right arm. The IRES activity of the mutant was also reduced to 21% of WT when the mutant was inserted into two pairs of bases (^446^CC^447^-^643^GG^644^) (mut11), while the mutant with an insertion of a pair of bases at 446 and 644 (CG) (mut12) retained 53% of WT IRES activity (Figure 4B). Besides these mutants within SLI, we also investigated the IRES activities located in the left arm next to the stem-loop I (mut 13–15) but found no significant differences compared to WT IRES activity (Figure 4C). Thus, these data suggest that specific regions within the stem-loop I affect the translation initiation activity of SVA IRES. 

To further study the effect of stem-loop I on initiating transcription in a more relevant cell type, the IRES activities of the fifteen mutants focused on SLI were also evaluated in porcine-derived ST cells. Although the results obtained for the IRES activities of these mutants were more variable in ST cells than in BHK-21 cells, a general correlation trend between both cell types could be made (Figure 4D,E). These experiments confirmed that mutating specific nucleotides within the stem-loop I affect the translation–initiation activity of SVA IRES in different cell lines. 

### 3.5. Mutations in Stem-Loop I Affect the Recovery of SVA In Vitro

To investigate the replication capacity of the mutant viruses with lower IRES activities, twelve full-length cDNA clones with mutations in the stem-loop I of SVA were constructed and transfected into BHK-21 cells. The mutant viruses harboring IRES sequences previously shown to have higher activities using the replicon systems showed relatively higher CPE in BHK-21 cells after transfection, whereas no CPE was observed after two or three generations for the mutants carrying the less efficient IRES sequences. Not surprisingly, the mut 9 and mut 11, which were only ~20% as efficient relative to the WT in BHK-21 and ST cells, failed to be rescued. Based on the IRES activity of the above-rescued viruses, the two mutants with higher IRES activity, L-C442U/U647C and S-ins C/G, and five mutants with lower IRES activity, L-del C/U, L-C442A, L-U443A, L-U646A and S-del C/G, were selected for growth kinetics analysis. Surprisingly, the characterization of the mutants revealed that all seven mutants had similar growth characteristics as the WT virus in both BHK-21 cells and porcine ST cells (Figure 5A). In addition, none of the mutants displayed any significant differences in total RNA levels when compared to the WT virus (Figure 5B). Thus, although some mutations in IRES stem-loop I did not significantly affect the SVA replication capacity, some mutations in stem-loop I affected the recovery of SVA, likely reflecting their essential roles for the initial round of translation required to start a successful infection.

## 4. Discussion

The IRES of the extensively studied flavivirus hepatitis C virus facilitates translation initiation of the viral ORF in a 5′ cap-independent manner [15]. Although from a completely different taxonomical classification, the IRES from the picornavirus SVA shares a high structural similarity to that of HCV. While IRES elements were shown to be responsible for the virulence phenotypes of several picornaviruses [28,29], a characterization of the *Senecavirus* genus IRES remains largely unexplored. By studying relationships between structure and function studies of the SVA IRES, we found here that the replacement of the SVA IRES by the intact HCV IRES impaired SVA replication in a cell type–specific manner (Figure 1C). Notably, the results indicated that that this is largely due to a decrease in IRES-dependent translation initiation efficiency. Additionally, fine mapping and characterization of key nucleotides within RNA structures demonstrated that the stem-loop I of SVA IRES is required for translational activity of the native SVA IRES, and that this domain is essential for the recovery of recombinant mutant viruses from cDNA clones.

Previously, we found that a critical virulence determinant on domain IV of the FMDV IRES through replacement of the natural IRES with its counterpart from bovine rhinitis B virus (BRBV) [24]. Additionally, a nucleotide substitution of cytosine (C) for guanine (G) at position 351 of the IRES endows FMDV with temperature-sensitive and attenuation phenotypes [25], highlighting the power of IRES manipulation as a feasible strategy to develop live-attenuated FMDV vaccines. With this motivation, we constructed a chimeric SVA virus in which the native IRES was entirely replaced with that of HCV. The chimeric virus exhibited an impaired growth phenotype in porcine-derived ST cells, but retained replication capacity in the hamster-derived BHK-21 cells (Figure 1C,D). Such a result was similar to that of our previous report on FMDV, which indicates that the IRES substitution in the 5’ UTR influences the replication efficiency of SVA in a cell-specific manner [24]. This finding prompted us to further characterize the role of the functional domain(s) of the SVA IRES in two different cell types.

Based on the secondary structure of chimeric IRES predicted in silico by the Mfold software, we constructed three IRES-chimeric SVA mutants in which each domain of HCV IRES was replaced with its counterpart from HCV. The results showed that the replacement of domain IV of SVA IRES with the corresponding domain of HCV had extremely low translation initiation activity and was lethal for SVA replication (Figure 2D). Previous studies have shown that the base pairing of IRES stems for HCV and CSFV are key for the translational activity of IRES and viral replication [30,31]. Therefore, we dug deeper into these domains and carried out further nucleotide substitutions within the IRES domain. As expected, only when both sides of the stem-loop I were complementary paired could the SVA IRES mutants be rescued. Importantly, the mutants that could not successfully be rescued were those with the lowest IRES translation–initiation activities (data not shown). In combination with the previously published data derived from other IRES-chimeric picornaviruses [32,33], our results for SVA indicate that the complementary base pairing of stem-loop I is indispensable for the IRES-mediated translation and virus recovery.

We also constructed a serial of plasmid-based SVA mutants, including deletion, insertion, transversion nucleotides within the stem-loop I, to explore which single or combined point mutations are essential for the activity of the SVA IRES. The dual-luciferase reporter assay in this study demonstrated that the IRES activities of these mutants in ST cells decreased at varying degrees (Figure 4E). Previously, Berry et al. constructed a set of mutants to structurally and mechanistically probe how the pseudoknot contributes to the activity of the HCV IRES, showing that the highly unstable structure of the pseudoknot caused by deleting two nucleotides in stem-loop I of the HCV IRES element could lead to inactivation of IRES [15]. Thus, we speculated that the disruption of the IRES structure within the stem-loop I would result in a decrease in the translational activity of the SVA IRES. The picornavirus IRES-driven translation initiation depends on the structural organization of the IRES and its interaction with cellular proteins, such as eIFs and ITAFs [12]. In addition, in vitro studies have identified different cell factors that bind to the different IRES structural domains for viral translation initiation [34,35]. Given what we know for other picornaviruses, the lower translational activity of SVA IRES mutants in this study may be due to a lack of structural stability of the IRES, which in turn could affect the interaction between the 5’ UTR and cell factors, leading to impaired IRES activity and virus replication. However, the specific mechanisms for these differences remain unknown and will be subjected to further studies.

Previous studies have reported that changes in IRES could also lead to picornavirus attenuation [27]. For example, a nucleotide mutation of C to U at position 472 of the poliovirus type 3 IRES endows neurovirulence phenotypes in primate and murine models [36,37]. The present study also indicated that inefficient translation is the major cause of the replication block of the IRES replaced (H-SVA-IRES) or domain-specific chimeric viruses. Unexpectedly, however, SVA mutants in the IRES Stem-loop I with lower IRES initial translation activity in vitro showed no discernable replication defects in either hamster-derived BHK-21 cells or porcine-derived ST cells (Figure 5A,B). Interestingly, motif mutations in the pseudoknot stem I upstream of the start codon in the SVA genome impacts IRES activity and the virus recovery [38]. Given that the data from bicistronic reporter systems only reflect IRES activity to some extent and are missing the context of other genetic elements present in the complete viral genome, we cannot rule out that other regions, such as the 3’ UTR, could be responsible for the discrepancies in our results [39]. For example, studies have identified that the 3’UTR, composed of two stem-loops and a poly(A) tract, is required for FMDV infectivity and stimulates IRES activity [40,41]. Therefore, we speculate that the lack of a quantifiable phenotype in the stem-loop I mutant viruses with impaired IRES activity is being masked by interactions between the SVA 3’ UTR and IRES. Future studies using the mutants created here with additional mutations in other genomic regions will address this possibility.

## 5. Conclusions

In summary, the results showed that the replacement of SVA IRES from HCV IRES leads to a decrease in the IRES translation–initiation activity and the viral replication in porcine-derived cells. A series of IRES-chimeric viruses were constructed in which each domain of the SVA IRES was replaced with its counterpart from the HCV IRES, demonstrating that the stem-loop I within domain IV of the IRES element is required for SVA IRES activity and viral recovery.

## Figures and Tables

**Figure 1 viruses-13-02159-f001:**
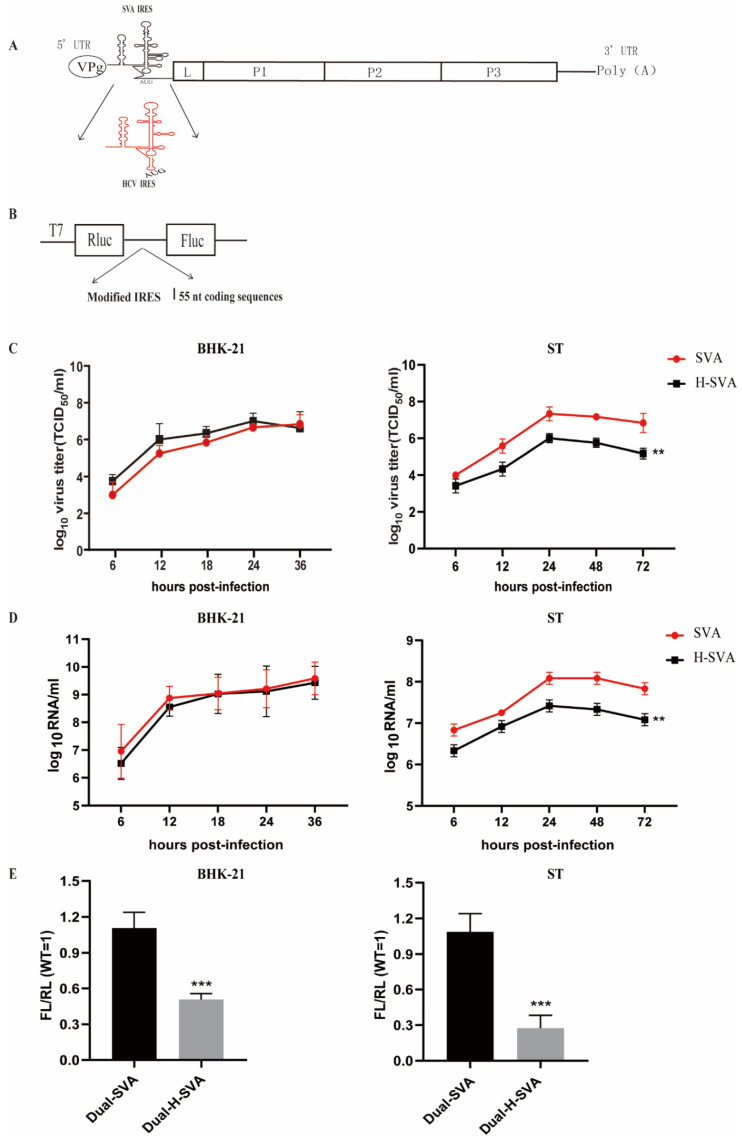
Construction and characterization of WT SVA and the IRES-replaced chimeric virus H-SVA. (**A**) Schematic diagram of the construction of SVA IRES replaced by HCV IRES. The black graph depicts SVA genome. The SVA IRES and HCV IRES is indicated by black and red graph, respectively. BHK-21 and ST cells were infected with SVA or H-SVA at an MOI of 0.01. (**B**) Schematic diagram of the construction of a dual luminescent reporter plasmid. Growth curves were determined by TCID_50_ assay, (**C**) and the viral RNA copies were determined by RT-qPCR (**D**). (**E**) IRES activity of WT SVA and H-SVA. The dual-luciferase reporter plasmids for WT SVA or H-SVA RNA were transfected into BHK-21 or ST cells. At 12 h post-transfection, the RLuc and FLuc activities in the cell lysates were quantified. Data in panels B, C, E are presented as the mean ± SD (*n* = 3). Significance of the changes were analyzed by Student’s *t*-test and indicated by *** *p* < 0.001, ** *p* < 0.01.

**Figure 2 viruses-13-02159-f002:**
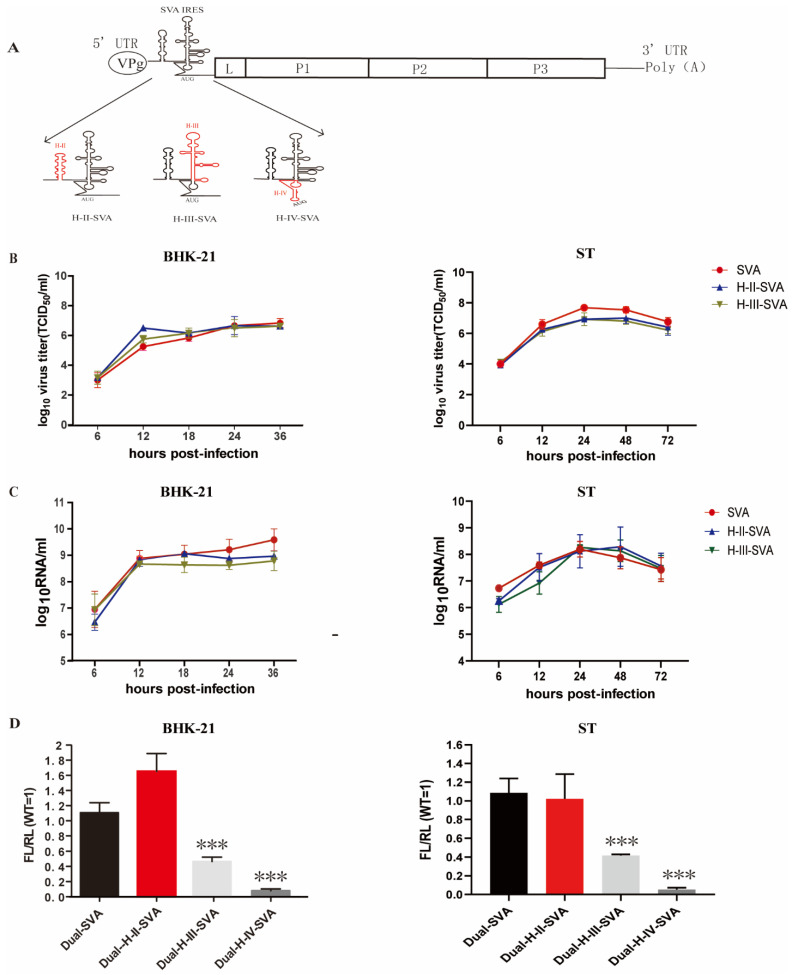
IRES activity and replicating ability of the domain specific IRES-chimeric SVA. (**A**) The schematic diagram for the construction of the IRES-chimeric SVA. The top black graph depicts SVA genome. The SVA IRES is indicated by arrows. The domain specific mutants used in this analysis are shown. The red sequences indicate replacement with HCV IRES counterparts. (**B**)BHK-21 and ST cells were infected with SVA or IRES-chimeric SVA viruses at an MOI of 0.01. Growth curves were determined by TCID_50_ assay. (**C**)The viral RNA levels of SVA or IRES-chimeric SVA were determined by RT-qPCR. (**D**) IRES activities of WT SVA and IRES-chimeric SVA; BHK-21 and ST cells were transfected with the dual-luciferase reporter plasmids of Dual-H-II-SVA, Dual-H-III-SVA, Dual-H-IV-SVA or Dual-SVA. At 12 h post-transfection, the RLuc and FLuc activities in the cell lysates were quantified. The results are presented as the means+SD from at least three independent experiments. Significance of the changes were analyzed by Student’s *t*-test and indicated by *** *p* < 0.001.

**Figure 3 viruses-13-02159-f003:**
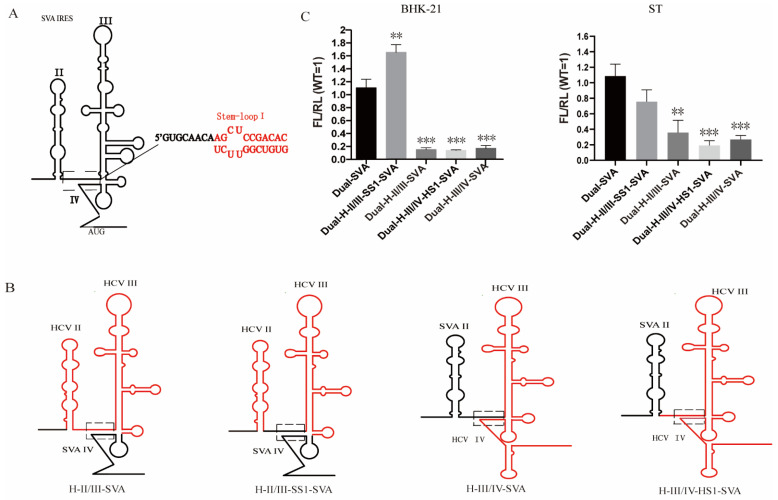
IRES activities of SVA chimera perturbing stem-loop I structure with replacement of double IRES domains. (**A**) The secondary structure of IRES stem-loop I of SVA; (**B**) The secondary structure of stem-loop I of the IRES-chimeric SVA with replacement of domains II and III or domains III and IV. The red and black regions represent the replacement of the HCV IRES and SVA IRES, respectively (**C**) IRES activities of IRES-chimeric SVA with replacement of double IRES domains. Data in panel C are presented as the mean ± SD (*n* = 3). Significance of the changes were analyzed by Student’s *t*-test and indicated by *** *p* < 0.001, ** *p* < 0.01.

**Figure 4 viruses-13-02159-f004:**
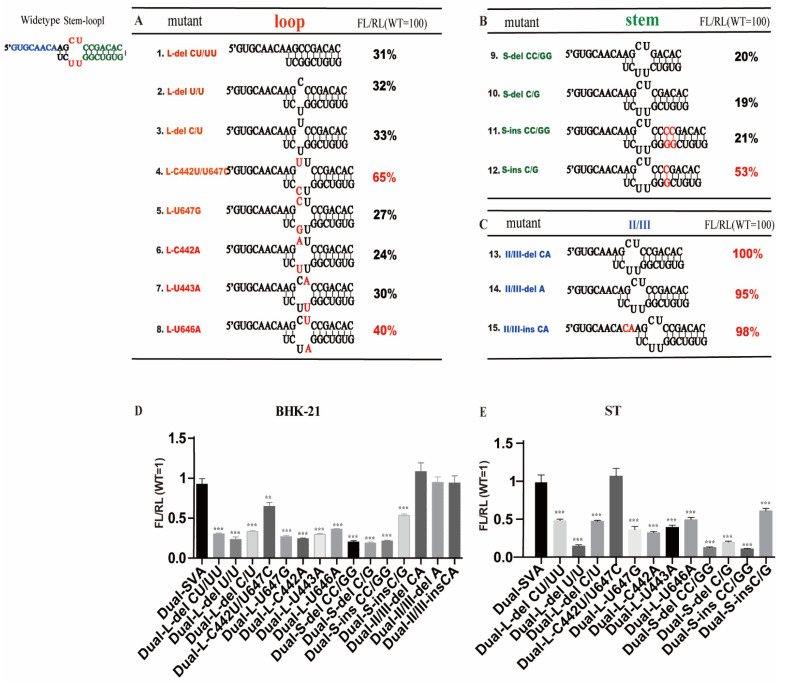
Mutational analysis of stem loop I. IRES activity with mutations in stem (**A**), loop (**B**) and inter-domain II/III region (**C**) IRES activity assay focused on the stem-loop I by mutations for further analyses. Monolayers of BHK-21 were transfected with in vitro-transcribed RNA of luciferase replicons and incubated at 37 °C. The activities of luciferase (Rluc) and Firefly luciferase (Fluc) were measured at 12 h post-transfection in BHK-21 (**D**) and ST cells (**E**). Data are means +SD from three experiments. Significance of the changes were analyzed by Student’s *t*-test and indicated by *** *p* < 0.001, ** *p* < 0.01.

**Figure 5 viruses-13-02159-f005:**
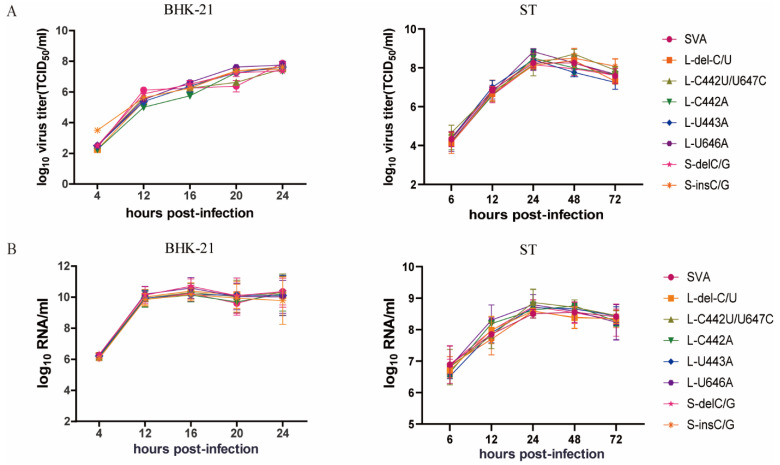
Replicating ability of SVA stem-loop I mutants. BHK-21 and ST cells were infected with mutants at an MOI of 0.01. The viruses produced were harvested at different times, and virus titers were determined as TCID_50_ /mL in BHK-21 cells and ST cells. Growth curves (**A**) and viral RNA loading (**B**) of mutants or SVA are shown. Error bars represent +SD (n = 3).

**Table 1 viruses-13-02159-t001:** Primers used for constructing SVA mutants.

#	Primer	Sequence(5′-3′)
1	NheI-F	GCT AGC ACT AGT TAA TAC GAC TCA CTA TAG GGT GTT AAG CG
2	SacII-R	AGT GTT TGC GTA GTA ATT GAA GGT CAT GTT ACC ATT ATT G
3	SacII-S1-R	TAA GTG TTT GCG TAG TAA TTG AAG GTC ATG TTA CCA TTA TTG
4	H-SVA-S-R	AGT AGT TCC TCA CAG GTC GCA GCC TTA AAA GGG ACT AAC AGC ATG TGG
5	H-SVA-H-F	CCT TTT AAG GCT GCG ACC TGT GAG GAA CTA CTG TCT TCA CGC AGA AAG
6	H-SVA-H-R	AAT GAG AGT TCT GCA TGG TGC ACG GTC TAC GAG ACC TCC CG
7	H-SVA-S-F	GTA GAC CGT GCA CCA TGC AGA ACT CTC ATT TTT CTT TCG ATA CAG CCT TG G
8	H-II-SVA-R	GAC TCT GTG TCG GAG CTT GTT GCA ACT GGA GGC TGC ACG ACA CTC ATA CTA ACG CCA TG
9	H-II-SVA-F	AGT GTC GTG CAG CCT CCA GTT GCA ACA AGC TCC GAC ACA GAG TCC ACG TGA TTG CTA CC
10	H-III-SVA-S-R	GGT TCC GCA GAC CAC TAT GGC TCT GTG TCG GAG CTT GTT GCA AGA AGG CCT CTC GGT TC
11	H-III-SVA-H-F	GGC CTT CTT GCA ACA AGC TCC GAC ACA GAG CCA TAG TGG TCT GCG GAA CCG GTG AGT AC
12	H-III-SVA-H-R	CCG ACA CGA CTA GGC CGT CAC CCT ATC AGG CAG TAC CAC AAG GCC TTT CGC GAC CCA AC
13	H-III-SVA-S-F	TGT GGT ACT GCC TGA TAG GGT GAC GGC CTA GTC GTG TCG GTT CTA TAG GTA GCA CAT AC
14	H-IV-SVA-R	CTC CCG GGG CAC TCG CAA GCG CCC TAT CAG GCA GTA TCC AAG GCA CGC TAA GGC CTA GC
15	H-IV-SVA-F	TAC TGC CTG ATA GGG CGC TTG CGA GTG CCC CGG GAG GTC TCG TAG ACC GTG CAC CAT GC
16	L-delCU/UU-ΔCU-R	GTA GCA ATC ACG TGG ACT CTG TGT CGG CTT GTT GCA AGA AGG CCT CTC GG
17	L-delCU/UU-ΔCU-F	CCG AGA GGC CTT CTT GCA ACA AGC CGA CACAGA GTC CAC GTG ATT GCT AC
18	L-delCU/UU-ΔUU-R	ATA TTT GTA TGT GCT ACC TAT AGC CGA CAC GAC TAG GCC GTC GCC CTA TC
19	L-delCU/UU-ΔUU-F	ATA GGG CGA CGG CCT AGT CGT GTC GGC TAT AGG TAG CAC ATA CAA ATA TG
20	L-delU/U-ΔU1-R	TAG CAA TCA CGT GGA CTC TGT GTC GGG CTT GTT GCA AGA AGG CCT CTC GG
21	L-delU/U-ΔU1-F	CGA GAG GCC TTC TTG CAA CAA GCC CGA CACAGA GTC CAC GTG ATT GCT AC
22	L-delU/U-ΔU2-R	TAT TTG TAT GTG CTA CCT ATA GAC CGA CAC GAC TAG GCC GTC GCC CTA TC
23	L-delU/U-ΔU2-F	TAG GGC GAC GGC CTA GTC GTG TCG GTC TAT AGG TAG CAC ATA CAA ATA TG
24	L-C442U/U647C-C442U-R	AGC AAT CAC GTG GAC TCT GTG TCG GAA CTT GTT GCA AGA AGG CCT CTC GG
25	L-C442U/U647C-C442U-F	GAG AGG CCT TCT TGC AAC AAG TTC CGA CACAGA GTC CAC GTG ATT GCT AC
26	L-C442U/U647C-U647C-R	GCA TAT TTG TAT GTG CTA CCT ATA GGA CCG ACA CGA CTA GGC CGT CGC CCT ATC
27	L-C442U/U647C-U647C-F	TAG GGC GAC GGC CTA GTC GTG TCG GTC CTA TAG GTA GCA CAT ACA AAT ATG CAG
28	L-U647G-R	AGC AAT CAC GTG GAC TCT GTG TCG GAT CTT GTT GCA AGA AGG CCT CTC GG
29	L-U647G-F	AGA GGC CTT CTT GCA ACA AGA TCC GAC ACA GAG TCC ACG TGA TTG CTA CC
30	L-C442A-R	AGC AAT CAC GTG GAC TCT GTG TCG GAT CTT GTT GCA AGA AGG CCT CTC GG
31	L-C442A-F	AGA GGC CTT CTT GCA ACA AGA TCC GAC ACA GAG TCC ACG TGA TTG CTA CC
32	L-U443A-R	AGC AAT CAC GTG GAC TCT GTG TCG GTG CTT GTT GCA AGA AGG CCT CTC GG
33	L-U443A-F	ACC GAG AGG CCT TCT TGC AAC AAG CAC CGACAC AGA GTC CAC GTG ATT GCT ACC
34	L-U646A-R	ATT TGT ATG TGC TAC CTA TAG ATC CGA CAC GAC TAG GCC GTC GCC CTA TC
35	L-U646A-F	GGG CGA CGG CCT AGT CGT GTC GGA TCT ATA GGT AGC ACA TAC AAA TAT GC
36	S-delCC/GG-ΔCC-R	GTA GCA ATC ACG TGG ACT CTG TGT CAG CTT GTT GCA AGA AGG CCT CTC GG
37	S-delCC/GG-ΔCC-F	CCG AGA GGC CTT CTT GCA ACA AGC TGA CAC AGA GTC CAC GTG ATT GCT AC
38	S-delCC/GG-ΔGG-R	GCA TAT TTG TAT GTG CTA CCT ATA GAA GAC ACG ACT AGG CCG TCG CCC TAT CAG
39	S-delCC/GG-ΔGG-F	CTG ATA GGG CGA CGG CCT AGT CGT GTC TTC TAT AGG TAG CAC ATA CAA ATA TGC
40	S-delC/G-ΔC-R	CAA TCA CGT GGA CTC TGT GTC GAG CTT GTT GCA AGA AGG CCT CTC GGT TC
41	S-delC/G-ΔC-F	CGA GAG GCC TTC TTG CAA CAA GCT CGA CAC AGA GTC CAC GTG ATT GCT ACC ACC
42	S-delC/G-ΔG-R	TAT TTG TAT GTG CTA CCT ATA GAA CGA CAC GAC TAG GCC GTC GCC CTA TC
43	S-delC/G-ΔG-F	TGA TAG GGC GAC GGC CTA GTC GTG TCG TTC TAT AGG TAG CAC ATA CAA ATA TGC
44	S-insCC/GG-insCC-R	GTA GCA ATC ACG TGG ACT CTG TGT CGG GGA GCT TGT TGC AAG AAG GCC TCT CGG
45	S-insCC/GG-insCC-F	CCG AGA GGC CTT CTT GCA ACA AGC TCC CCG ACA CAG AGT CCA CGT GAT TGC TAC
46	S-insCC/GG-insGG-R	ATA TTT GTA TGT GCT ACC TAT AGA ACC CCG ACA CGA CTA GGC CGT CGC CCT ATC
47	S-insCC/GG-insGG-F	ATA GGG CGA CGG CCT AGT CGT GTC GGG GTT CTA TAG GTA GCA CAT ACA AAT ATG
48	S-insC/G-insC-R	GTG GTA GCA ATC ACG TGG ACT CTG TGT CGG GAG CTT GTT GCA AGA AGG CCT CTC
49	S-insC/G-insC-F	AGA GGC CTT CTT GCA ACA AGC TCC CGA CAC AGA GTC CAC GTG ATT GCT ACC ACC
50	S-insC/G-insG-R	TAT TTG TAT GTG CTA CCT ATA GAA CCC GAC ACG ACT AGG CCG TCG CCC TAT CAG
51	S-insC/G-insG-F	GAT AGG GCG ACG GCC TAG TCG TGT CGG GTT CTA TAG GTA GCA CAT ACA AAT ATG
52	II/III-delCA-R	ATC ACG TGG ACT CTG TGT CGG AGC TGT TGC AAG AAG GCC TCT CGG TTC CC
53	II/III-delCA-F	CTT AGT AAG GGA ACC GAG AGG CCT TCT TGCAAC AGC TCC GAC ACA GAG TCC ACG
54	II/III-delA-R	AAT CAC GTG GAC TCT GTG TCG GAG CTT TGC AAG AAG GCC TCT CGG TTC CCT TAC
55	II/III-delA-F	TAA GGG AAC CGA GAG GCC TTC TTG CAA AGCTCC GAC ACA GAG TCC ACG TGA TTG
56	II/III-insCA-R	GTG GAC TCT GTG TCG GAG CTT GTG TTG CAA GAA GGC CTC TCG GTT CCC TTA C
57	II/III-insCA-F	AGT AAG GGA ACC GAG AGG CCT TCT TGC AACACA AGC TCC GAC ACA GAG TCC ACG

**Table 2 viruses-13-02159-t002:** Primers used for construction of bicistronic reporter plasmids of SVA.

#	Primer	Sequence (5′-3′)
1	Dual-SVA-B/N-F	GCC GTG TAA GAA TTC GAA GAT CTG ATG GCT ATC CAC
2	Dual-B/N-R	CGA AGT CAT GGA TCC CAT GGC ATG GTT ACG TCT
3	Dual-HCV-B/N-F	CCC GGG CTC GAG ATC TCC TGT GAG GAA CTA CTG TC

## Data Availability

The data presented in this study are available on request from the corresponding author.

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
