# Peer review of "The Stem-Loop I of Senecavirus A IRES Is Essential for Cap-Independent Translation Activity and Virus Recovery"

_viruses, 2021, doi:10.3390/v13112159_

Round 1

Reviewer 1 Report

The manuscript was well written, the experiments were well-conducted, the description of the results is clear, and the discussion section clearly describes the results and limitations. I have no significant comments on this work.

I have only two minor comments.

 It is not clear that Figure 1 A represents In silico predictions of the RNA secondary structure of the SVA IRES and the HCV. Figure legend has a different description.

I think the authors can increase the figure quality. 

Reviewer 2 Report

This is an interesting manuscript describing a genetic engineering studies on Senecavirus A causing vesicular disease in swine. The authors confirmed  the similarity of IRES region to other viruses represented by hepatitis C virus (HCV). Having a genetically modified variants of Senecavirus the authors revealed that chimeric SVA virus harboring the IRES from HCV, H-SVA, is viable and replicated normally in rodent-derived BHK-21 cells but showed replication defects in porcine-derived ST cells. The authors showed that the  stem-loop I of SVA is an essential element for IRES-dependent translation  and viral replication. 

I would strongly suggest to limit the overuse of personal form throughout the entire manuscript. Probably it would be possible to introduce non-personal form: 'it has been found' or 'the results showed' instead of "we showed" etc.
